# Expression of Selected miRNAs in Normal and Cancer-Associated Fibroblasts and in BxPc3 and MIA PaCa-2 Cell Lines of Pancreatic Ductal Adenocarcinoma

**DOI:** 10.3390/ijms24043617

**Published:** 2023-02-10

**Authors:** Václav Mandys, Alexey Popov, Robert Gürlich, Jan Havránek, Lucie Pfeiferová, Michal Kolář, Jana Vránová, Karel Smetana, Lukáš Lacina, Pavol Szabo

**Affiliations:** 1Department of Pathology, Third Faculty of Medicine, Charles University and University Hospital Královské Vinohrady, 100 00 Prague, Czech Republic; 2Department of Surgery, Third Faculty of Medicine, Charles University and University Hospital Královské Vinohrady, 100 00 Prague, Czech Republic; 3Institute of Molecular Genetics, Czech Academy of Sciences, 100 00 Prague, Czech Republic; 4Laboratory of Informatics and Chemistry, University of Chemistry and Technology, 166 28 Prague, Czech Republic; 5Department of Medical Biophysics and Medical Informatics, Third Faculty of Medicine, Charles University, 100 00 Prague, Czech Republic; 6First Faculty of Medicine, BIOCEV, Charles University, 252 50 Vestec, Czech Republic; 7First Faculty of Medicine, Institute of Anatomy, Charles University, 128 00 Prague, Czech Republic; 8Department Dermatovenereology, First Faculty of Medicine, Charles University and General University Hospital, 128 08 Prague, Czech Republic

**Keywords:** pancreas, cancer-associated fibroblast, miRNA, miR-21, miR-210, IL-6, hypoxia

## Abstract

Therapy for pancreatic ductal adenocarcinoma remains challenging, and the chances of a complete cure are very limited. As in other types of cancer, the expression and role of miRNAs in controlling the biological properties of this type of tumor have been extensively studied. A better insight into miRNA biology seems critical to refining diagnostics and improving their therapeutic potential. In this study, we focused on the expression of miR-21, -96, -196a, -210, and -217 in normal fibroblasts, cancer-associated fibroblasts prepared from a ductal adenocarcinoma of the pancreas, and pancreatic carcinoma cell lines. We compared these data with miRNAs in homogenates of paraffin-embedded sections from normal pancreatic tissues. In cancer-associated fibroblasts and cancer cell lines, miRNAs differed significantly from the normal tissue. In detail, miR-21 and -210 were significantly upregulated, while miR-217 was downregulated. Similar transcription profiles were earlier reported in cancer-associated fibroblasts exposed to hypoxia. However, the cells in our study were cultured under normoxic conditions. We also noted a relation to IL-6 production. In conclusion, cultured cancer-associated fibroblasts and carcinoma cells reflect miR-21 and -210 expression similarly to the cancer tissue samples harvested from the patients.

## 1. Introduction

Pancreatic ductal adenocarcinoma (PDAC) is a relatively infrequent disease with a reported incidence of more than 64,000 cases in the USA in 2021. Unfortunately, the options for efficient therapy are currently limited, and patient prognosis remains abysmal [1]. A typical histological feature of this type of cancer is stromal desmoplasia. The scar-like stroma contains a copious extracellular matrix produced by cancer-associated fibroblasts (CAFs).

The structural roles of CAFs include the production and simultaneous remodelling of the extracellular matrix [2,3]. Concerning the proportion of distinct cell types present in PDAC tumors, CAFs represent an outstandingly abundant cell type. The surprising quantity of CAFs in PDAC can be easily compared to the number of malignant cells in other types of carcinomas. Regarding their origin, CAFs can arise from local fibroblasts. However, their origin in mesenchymal stem cells, pericytes, or stellate cells under the control of cancer cells has also been observed [3].

The stroma is biologically highly active and significantly influences tumor growth, metastatic spread, and resistance to therapy [4]. Therefore, it is likely that CAFs influence the biology of PDAC through several mechanisms [5,6,7]. Primarily, CAFs influence malignant cells across a plethora of tumor types [8], including PDAC, by their paracrine production. Secreted growth factors, cytokines, and chemokines constitute a chronically inflammation-supporting tissue microenvironment. CAFs can actively influence cancer cells; however, various immune cell types are also present in the tumor microenvironment. The anti-tumor immunity can become inefficient in this specific microenvironment or even reverse its function to foster malignant cell growth [9].

This can be further enhanced by exosomes bearing cargos of bioactive proteins and mRNAs. The cancer-supporting properties of CAFs are significantly stimulated by exosomes produced by cancer cells containing numerous miRNAs as bioactive cargo [3,10]. Almost 3000 different miRNAs, including miR-21 and -210, have been detected in exosomes so far and listed in publicly accessible databases (http://exocarta.org/, accessed on 27 October 2022).

The small non-coding RNAs called miRNAs are involved in numerous regulatory processes in normal and cancer cells. miRNAs can become particularly important in mutual cross-talk between cancer cells and other elements forming the cancer ecosystem. Mechanistically, miRNAs can interact with mRNAs and so influence the process of translation. This may reduce specific protein production as the final step of execution of the gene program [11,12]. This post-transcriptional mechanism ensures control over key cellular processes such as proliferation, migration, and apoptosis, representing fundamental properties of cancer growth, spread, and resistance to therapy. Because of the poor prognosis of PDAC treated by recently available drugs, the regulatory processes mediated by miRNAs are being extensively studied in this type of malignancy as a potential target for novel therapeutic interventions [13,14,15]. Before the animal or even clinical stages of experiments may be initiated, we have to gain deeper insight into the regulatory pathways using in vitro models of PDAC.

In this study, we aimed to produce PDAC models in vitro using cultured primary normal and cancer-associated fibroblasts prepared from the PDAC (PANF), together with two well-characterized PDAC cell lines, BxPC3 and MIAPaCa-2.

We first addressed the expression of five selected miRNAs (miR-21, miR-96, miR-196a, miR-210, and miR-217) in normal pancreatic tissues. In closer detail, we selected these five miRNAs because they play a significant role in PDAC concerning tumor morphology, progression, and clinical outcomes. Notably, miR-21 and miR-210 proved to have a certain diagnostic/prognostic value [14,16,17]. Our previous work addressed the expression profiles of normal fibroblasts and PANF at the whole-genome scale [5]. In this study, we focus on miRNA detection with emphasis on genes that have a functional relevance to miRNA expression and thus to the biology of PDAC. In parallel, we also focus on cytokine IL-6, a critically important master regulator of the interplay between cancer cells and non-malignant cells within the cancer ecosystem [3]. Thus, the main purpose of this article is to provide a basis for comparison of the data obtained from in vitro experiments with the results of studies on clinical material obtained from PDAC patients.

## 2. Results and Discussion

### 2.1. Immunocytochemical Characterisation of HFs and PANF

While both studied fibroblasts types, i.e., human fibroblasts (HFs) and PANF, are rich in vimentin, only PANF cultures contain a high proportion of myofibroblasts (Figure 1). Using immunocytochemical detection, we observed a high percentage of cells (close to 100%) expressing α-smooth muscle actin with variable intensity. In unstimulated cultures of primary HFs, we observed only scarcely distributed positive cells. In many of these positive PANF cells, we observed a highly organised actin cytoskeleton, typical of fully differentiated myofibroblasts [18].

This is also typical of CAFs [3], which significantly influence the functional properties of PDAC malignant cells [5]. At the mRNA level, the expression of the *ACTA2* gene for α-smooth muscle actin in PANF increased less than two-fold (fold change = 1.5) with marginal statistical significance (adjusted *p*-value = 0.06) and was thus considered non-significant in contrast to *ACTG2* encoding γ-smooth muscle actin, which increased both more than two-fold and was statistically significant (Appendix A). Both genes *ACTA2* and *ACTG2* encode smooth muscle actin. *ACTA2* is the gene encoding α-smooth muscle actin expressed, e.g., in the vascular wall. However, it is also expressed in fibroblasts, myofibroblasts, and smooth muscle cells elsewhere. *ACTG2* is the gene encoding γ-smooth muscle actin, also known as the enteric form for its occurrence mainly in smooth muscle cells of the intestine [Human Protein Atlas, https://www.proteinatlas.org/, accessed 11 January 2023]. *ACTA2* and *ACTG2* are highly homologous, and the regional distribution of expression is not mutually exclusive. Although the commercial antibody employed for detecting SMA was designed to recognize α-smooth muscle actin, because of the similarity between both actin molecules, the interaction with γ-smooth muscle actin cannot be excluded. Co-expression of both isoforms in gastrointestinal cancer has also been reported by others [19].

### 2.2. miRNA Expression

Our results demonstrated that the expression of both miR-21 and miR-210 in cultured cells was significantly higher than in the formaldehyde-fixed paraffin-embedded tissue from the normal pancreas (Figure 2). MicroRNA miR-217 was significantly downregulated (up to 100-fold) (Figure 3). The expression of miR-96 and miR-196a did not differ from the normal pancreas tissue in a consistent way (Figure 4, Appendix A).

To provide robust controls for our cell-based research, we built our experiments on two independent biological replicates of fibroblasts. Interestingly, both types of normal fibroblasts used in our study significantly differed in the expression of all of the studied miRNAs except for miR-210 (Figure 2, Figure 3 and Figure 4). This surprising difference can be explained by their different embryonic origin. Facial-skin-derived HDF fibroblasts originated in ectomesenchyme derived from the neural crest. The truncal-skin-derived HFs originated from the mesoderm [20]. This developmental difference must be taken into account cautiously. The different origin of these types of normal fibroblasts has a significant effect on their expression profile and presumably a significant influence on their function [21,22]. We therefore prefer to present both fibroblast controls separately and evaluate the statistical significance independently where relevant (Figure 2 and Figure 3).

Furthermore, it is not easy to explain the observed difference in the production of both miR-21 and miR-210 between BxPc3 and MIA PaCa-2. Various phenotypic differences between both cell lines have been published earlier by others. While MIA PaCa-2 cells have a typical epithelioid morphology, BxPc3 cells have a mesenchymal morphology. BxPc3 cells are, moreover, highly resistant to chemotherapy [23].

Both miR-21 and miR-210 are known to be activated by hypoxic conditions. Hypoxia occurs in cancer, during wound healing, as well as in other pathological conditions [24,25,26,27,28]. Lowering miR-21 and miR-210 expression decreases cell migration and invasion of pancreatic stellate cells (PSCs) and CAFs [29]. Predominantly, miR-21 can control various functions of fibroblasts, such as extracellular matrix production, angiogenesis, and inflammation control. This can be particularly important during wound healing and cancer progression [30]. This striking similarity is not surprising, because wound healing and the cancer microenvironment exhibit many similarities at the cellular and molecular levels [31,32].

We also observed this significant downregulation of miR-217 in cultured cells, mainly in PANF and cancer cells (Figure 3).

miR-217 is a senescence-associated miRNA and it can induce cellular senescence in normal fibroblasts [33]. On the other hand, miR-217 is usually expressed in the normal pancreas [34]. A dual-luciferase reporter assay revealed that KRAS mRNA is the direct target of miR-217. Overexpression of miR-217 in a PDAC cell line decreases KRAS mRNA levels and inhibits cell proliferation [35]. These findings can possibly explain the low expression levels of miR-217 that we observed in fibroblasts and pancreatic cancer cell lines. Moreover, downregulation of miR-217 may predict cancer presence in PDAC patients [36]. Downregulation of miR-217 (as well as miR-96) is known to be important for cancer cell proliferation and migration. This was also confirmed by the data obtained from PDAC [14,34,37].

Although two other detected miRNAs, i.e., miR-96 and miR-196a, in cultured cells also statistically differed from the normal pancreas, they were actually very close to the level detected in the normal tissue (Figure 4). Their difference from the normal tissue was lower than our two-fold cut-off [38] except in cancer cells and was somewhat inconsistent in our experiments.

However, the expression of miR-196a can also be potentially relevant to PDAC biology. This miRNA represents a component of the molecular signature of Hodgkin lymphoma [39], where it controls cancer cell proliferation, as confirmed in different types of tumors as well [40].

### 2.3. mRNA Expression in Normal Fibroblasts, CAFs, and PDAC Cell Lines

The CAFs used in this study were prepared from PDAC (PANF) or from cutaneous malignant melanoma (MELF). Together with normal dermal fibroblasts (HFs), all of the isolated primary cells were described in detail earlier [5]. Overall, we observed agreement between changes in the miRNA species and changes in the mRNA expression of their targets, albeit this association was not statistically significant. Targets of miR-196a and miR-96 also showed deregulation (Appendix A).

While comparing PANF and HFs, we identified 1327 differentially expressed transcripts (see Appendix A) belonging to many biological processes. This broad difference in expression is not surprising, as we compared CAFs from an internal organ with normal dermal fibroblasts. When studied at the level of biological processes using the gene set enrichment analysis based on the Gene Ontology terms, we observed differences in genes associated with organ development, vasculature development, and tissue morphogenesis, but also in response to extracellular matrix reorganisation and response to hypoxia (Appendix A).

The CAFs indeed expressed many factors typical of hypoxic conditions when compared to HFs, although they were cultured under the same conditions in the atmospheric pressure of oxygen (Figure 5, Appendix A). In PANF, this observation harmonises with the detected high expression of both miR-21 and miR-210 described above. Surprisingly, the crucial player of hypoxia signalling, HIF-1α (*HIF1A* gene), was not significantly upregulated in PANF (expression increased by 60% in PANF and with marginal statistical significance, adjusted *p*-value = 0.09). However, even this level of change in *HIF1A* gene activity was reflected by the expression intensity of its targets (Figure 5). Data from PANF were also compared with data from MELF, which were employed for an independent comparison to CAF fibroblasts. Interestingly, these CAFs differed from the normal fibroblasts but shared common features with PANF cells (Figure 5).

The expression of *HIF1A* in cooperation with both miR-21 and miR-210 can influence the properties of PDAC cells, such as proliferation, epithelial–mesenchymal transition, and migration [41]. The neovascularisation of the tumor is essential for its sustained growth and metastatic spread. The orchestration between the expression of miR-21, miR-210, *HIF1A*, and *VEGFA* (which is significantly upregulated in PANF) seems to strongly support PDAC vascularisation and progression [42]. Moreover, the *HIF1A*–*VEGFA* axis seems to be involved in the control of PDAC cell invasiveness [43]. Similarly, both miR-21 and miR-210 cooperate with carboanhydrase IX (*CAIX*), also expressed by PANF (Figure 6). Carboanhydrase IX is present in the acinar and ductal cells of both the normal pancreas and PDAC [44]. This protein has also been found in different types of cancer, namely cancer of the kidney [45]. The role of this enzyme in tumor vascularisation has also been reported by others [46].

These factors can also participate in the regulation of cyclin D2. The activity of the *CCND2* gene was significantly upregulated in PANF (Figure 6). This cyclin influences malignant cell proliferation [47]. Another hypoxia-dependent gene, *TP53I11*, was non-significantly upregulated in PANF as well (Figure 6). It is also under the influence of miR-210 and controls the susceptibility of cancer cells to cytotoxic T lymphocytes [48]. The interaction of miR-210 with the *BNIP3* transcript (significantly upregulated in PANF) can protect cancer cells against hypoxic damage [49]. miR-210 also regulates signalling cascades with the central position of the *TIMP1* gene [50] non-significantly upregulated in PANF. The TIMP1 protein is known as an inhibitor of metalloproteinases participating in the remodelling of the extracellular matrix in tumors [51]. Another deregulated gene family observed in this study were genes encoding members of the IGFBP protein family (Figure 6). Similar findings were also confirmed in malignant glioblastoma, where miR-21 inactivates IGFBP3. It consequently stimulates progression of this type of brain cancer [52].

### 2.4. Cross-Talk of miR-21 and IL6 Signalling in Hypoxia

CAFs represent an important component of the ecosystem in the majority of human malignant tumors. CAFs significantly influence clinically relevant aspects, such as cancer cell proliferation, maintenance of low differentiation status of cancer cells, and their capacity to form metastasis [53]. In addition to other molecules, CAFs express a panel of miRNAs influencing the described malignant cell behaviour. MiR-21 is highly expressed in normal dermal fibroblasts (HFs) and insignificantly in PANF. MiR-210 is significantly upregulated in PANF (and in MIA PaCa-2 cancer cells). Based on the literature, both these miRNAs significantly aggravate the patient’s prospect of survival in several types of malignant diseases, including PDAC (Table 1). MiR-21 was identified to distinguish different tumor subtypes in The Cancer Genome Atlas (TCGA) PAAD dataset [54], and high expression of both miR-21 and miR-210 worsened the patients’ survival in the same dataset (Appendix A).

CAFs and their products support the stable pro-inflammatory microenvironment that stimulates cancer cells and simultaneously abrogates the anti-tumor immune response [3,32]. Cytokine IL-6 seems to be in the central position of regulation of the crosstalk between cancer cells and non-cancerous cells in the tumor ecosystem [80,81]. The IL-6 role and activity was also confirmed in PANFs [5]. Both miR-21 and miR-210 seem to actively participate in the described phenomena. The upregulation of miR-21 in CAFs was observed after their targeting by exosomes with IL-6 cargo. This mechanism stimulates production of IL-6 in precancerous tissue of the uterine cervix [82], cutaneous malignant melanoma [83], and in colon cancer [84]. The interaction of miR-21 and miR-210 participates in the control of STAT3, the critical component of the IL-6 signalling pathway. Furthermore, collaboration with the effect on PI3K/AKT also influences IL-6 signalling, as observed in several types of cancer [85,86,87,88,89,90]. Interestingly, all of that was under hypoxic conditions.

The orchestration between miR-21 and miR-210 and the IL-6 signalling cascade was also noted in PDAC. This may have a therapeutic consequence concerning the reduction of tumor vascularization [24]. In this study, PANF with high expression of miR-21 and miR-210 exhibited higher expression of the *IL6* gene. However, the expression of the gene encoding the IL-6 receptor (*IL6R* gene) and the gene for signal transducer gp130 (*IL6ST*) was not significantly deregulated (Figure 6). Similarly, the expression of the gene encoding protein STAT3 was increased but did not reach significance (Figure 6). This observation supports the hypothesis about the effect of miR-21 and miR-210 on IL-6 production, but the sensitivity of these cells to IL-6 is not easy to predict.

Interestingly, a soluble form of IL-6R comprising the extracellular portion of the receptor can bind IL-6 with a similar affinity as the membrane-bound IL-6R. This soluble form of IL-6R can be secreted by other cell types and can even activate cells not expressing IL-6R at all. This process has been called trans-signalling and it demonstrates the remarkable versatility of the IL-6 signalling pathway. Via canonical signalling and trans-signalling together, IL-6 can orchestrate the complex ecosystem of a tumor composed of multiple cell lineages, eliciting individual responses in every one of them [91].

Another cytokine of the IL-6 family, leukaemia inhibitory factor (LIF), is also strongly upregulated in PANF, while only insignificantly increased in MELF, Figure 7. LIF is overexpressed in a variety of solid tumors, including pancreatic tumors [92], and promotes cancer cell proliferation [93,94]. This gene was found to be induced by hypoxia through HIF1A activation [95]. The LIF promoter region contains hypoxia-responsive elements that can be transcriptionally activated by hypoxia [94]. It upregulates miR-21 expression under various physiological and pathological conditions, e.g., during the maturation of oocytes into cumulus–oocyte complexes [96,97], in trophoblast cells [98] and in tumor cells [94].

PANF exhibited a significant difference in extracellular matrix organization compared to normal fibroblasts (Figure 8, Appendix A). The *PLOD2* gene encoding protein lysyl hydroxylase was found to be significantly upregulated in PANF. The product of this gene is critical for the crosslinking of collagen. It thus stabilizes the structure of the extracellular matrix (Figure 8, Appendix A). The high expression of PLOD2 is associated with poor prognosis in cancer patients [99,100]. Significant downregulation of MMP3 and MMP12 participating in the degradation of extracellular matrix molecules was also detected (Figure 8, Appendix A). As mentioned in the introduction, the stroma of PDAC is remarkably rich in the extracellular matrix, and it frequently exhibits desmoplastic features. These findings demonstrating differences in production and remodelling of ECM between HFs, MELF, and PANF harmonize well with the observation of expression of all of the differentially expressed miRNAs. The upregulation of miR-21 and miR-210 and the downregulation of miR-217 expression observed in this study seem to be linked with fibrosis under different pathological conditions, including desmoplastic stroma of PDAC [101,102,103,104]. This process is also related to hypoxic signalling [105,106] and is in agreement with the elevated expression of genes influenced by hypoxia (Figure 5).

### 2.5. Consequences of Upregulation of miR-21 and miR-210 and Downregulation of miR-217 by Cancer Cells and CAFs Prepared from Pancreatic Cancer

PANF isolated from PDAC maintained high levels of both miR-21 and -210 even after in vitro propagation. These data show that CAFs of PDAC can participate in their elevated levels in biological fluids, as observed by others [101,107]. It is of practical importance because both miRNAs represent a diagnostic marker of PDAC and have recently been proposed as prognostic markers [95,101]. They can even be used to distinguish PDAC from pancreatitis [108,109]. miR-21 and miR-210 upregulation and miR-217 downregulation were detected in CAFs and cancer cells prepared from PDAC [34]. The regulation by these miRNAs can influence the pro-tumorigenic microenvironment of PDAC.

## 3. Material and Methods

### 3.1. Cell Culture

HDFs isolated from facial skin were obtained from Cell Applications, Inc. (San Diego, CA, USA) and maintained in a fibroblast growth medium (Cell Applications, Inc., San Diego, CA, USA) supplemented with 10% foetal bovine serum (FBS) and ATB (penicillin 100 U/mL; streptomycin 100 μg/mL), both purchased from Sigma Aldrich Co (St. Louis, MO, USA). The other biological replicate, human fibroblasts (HFs), was isolated in our laboratory from the residual skin of a healthy donor undergoing routine aesthetic breast surgery at the Department of Aesthetic Surgery, Third Faculty of Medicine, Charles University. Cancer-associated fibroblasts from DACP (PANF) were prepared from the tumor samples obtained from the Department of Pathology, Third Faculty of Medicine, Charles University. HF and PANF isolation have been described elsewhere [110]. Both the HFs and PANF were also employed in our previous study [5], where both tested negative for epithelial markers (keratins), a leukocyte marker (CD45), an endothelial marker (CD34), and melanocytic markers (MiTF, HMB45, and MELAN-A). The non-commercial samples were obtained following informed consent from the patient with the agreement of the local ethics committee according to the Helsinki Declaration [111]. Pancreatic cell lines (BxPc3 and MIA PaCa-2) were obtained as a generous gift from Dr. Maurizio Viale, National Institute for Cancer Research, Genova, Italy. All of these lines were maintained in Dulbecco’s modified Eagle’s medium (DMEM) supplemented with 10% foetal bovine serum (FBS) and ATB (penicillin 100 U/mL; streptomycin 100 μg/mL), all purchased from Sigma Aldrich Co (St. Louis, MO, USA). The cells were seeded into 75 cm^2^ culture flasks. Nunclon was purchased from Thermo Fisher Scientific–Nunc A/S, Roskilde, Denmark (density 4 × 10^5^ cells/mL) and maintained in a 37 °C incubator with humidified air supplemented with 5% CO_2_. These cells were employed for the study of miRNAs.

### 3.2. Immunocytochemical Analysis

For immunocytochemical analysis, the cells were seeded at a density of 20,000/cm^2^. After 48 h, the cells were briefly washed with PBS and fixed in 2% paraformaldehyde. After permeabilization in TBS-T/1% hydrogen peroxide solution, the cells were blocked in 5% Roti-ImmunoBlock (CarlRoth, Karlsruhe, Germany). Primary antibodies (rabbit monoclonal anti-vimentin antibody [clone SP20] (ab16700), Abcam, Cambridge, UK) and mouse monoclonal anti-SMA [clone 1A4], Dako, Glostrup, Denmark) were diluted 1/100 in DAKO-Real antibody diluent and incubated overnight at 4 °C. After washing, the secondary polymer HRP-tagged antibody (Histofine^®^ Simple Stain™ M.A.X. PO MULTI; Nichirei Biosciences, Tokyo, Japan) was incubated for 30 min. Chromogenic detection was performed using Histofine^®^ Simple Stain™ A.E.C. Solution. The slides were counterstained in Gill’s hematoxylin and mounted in Biomount Aqua (Baria, Praha, Czech Republic).

### 3.3. MicroRNA Isolation and Reverse Transcription

After 120 h when the confluent growth of cells was approached, the cells were washed three times with a PBS solution, scraped with a rubber policeman, and harvested by centrifugation at 400× *g* for 5 min at room temperature. MicroRNAs were extracted from human cell lines with an miRNeasy kit (Qiagen, Hilden, Germany), following the manufacturer’s instructions. Formalin-fixed paraffin-embedded (FFPE) blocks with normal pancreatic tissue used as a negative control were retrieved from the archive of the Department of Pathology of the University Hospital Kralovske Vinohrady in Prague. This control sample (n = 1) was obtained from a patient undergoing partial pancreatectomy for a suspected tumor that was later histologically diagnosed as the accessory spleen [14]. The tissue was therefore considered normal and contained normal fibroblasts in contrast to those prepared from a pathological pancreas, which are often activated even when harvested far from the affected tissue [112]. The sample was routinely processed and embedded in paraffin, as described in [113]. Importantly, this type of preparation enables successful isolation of miRNAs and can be used for comparison with isolates from cultured cells [114,115].

One to three 6µm-thick unstained paraffin-embedded tissue sections were procured for miRNA extraction using an miRNeasy FFPE kit (Qiagen, Hilden, Germany), according to the manufacturer’s instructions. Reverse transcription was carried out using RevertAid Reverse Transcriptase (Thermo Fischer Scientific, Waltham, MA, USA) in a 50 μL reaction mixture containing the following reagents: 1 μg of DNA-free RNA, a reaction buffer [50 mM Tris-HCl (pH 8.3 at 25 °C), 50 mM KCl, 4 mM MgCl_2_ and 50 mM DTT], 1 mM of dATP, dTTP, dCTP, and dGTP, 20 IU rRNasin ribonuclease inhibitor, 100 IU of Moloney murine leukaemia virus reverse transcriptase (M-MuLV RT), and the primer mix, including 20 pmol of each stem-loop primer. A mix of stem-loop primers was used for miRNA reverse transcription. The primers were designed with miRNA primer designer software, kindly provided by Dr. Fuliang Xie, East Carolina University, USA. The stem-loop primer sequences for the alien spike (miR-39 from *C. elegans*) and the examined miRNAs are listed in Appendix A. Artificial spike RNA (miR-39 from *C. elegans*, 5 × 10^8^ copies) was also added to the reaction as an external reference. After initial denaturation (5 min at 70 °C, then cooling the samples on ice), the reactions were incubated at 25 °C (10 min), and then at 42 °C for 1 h. To stop the reaction, the mixture was heated at 70 °C for 10 min.

### 3.4. Real-Time qPCR

cDNA samples were amplified in duplicates using an Applied Biosystems 7500 Fast real-time PCR system and Hot FirePol EvaGreen qPCR Mix Plus (Solis BioDyne, Tartu Estonia). The reaction mix included 10 pmol of each primer (miRNA-specific and universal (Appendix A) and 2 μL of cDNA. Amplification of the cDNAs was performed under the following thermal conditions: denaturation at 94 °C for 15 min, followed by 40 cycles consisting of denaturation at 94 °C for 15 s, annealing at 62 °C for 60 s, and DNA synthesis at 72 °C for 40 s. The reaction product specificity was controlled with the respective melting curves. The ΔΔCt method was applied to measure the values of miRNA expression of interest [116] with spiked-in miR-39 from *C. elegans* used as a reference miRNA.

### 3.5. Comparison with the mRNA Expression Profile of PANF and CAFs from Melanoma (MELF) Based on [5]

The data were processed as described in the original article. In short, oligo [117] and limma [118] packages of R/Bioconductor [119] were used to identify differentially transcribed genes after the transcription profiles were background corrected using a normal–exponential model, quantile normalised, and variance stabilized using base 2 logarithmic transformation. A moderated *t*-test was used to detect differentially expressed transcripts. An adjusted *p*-value (Storey’s q) of <0.05 and a minimally two-fold change in expression intensity were required to consider the gene as differentially transcribed. The MIAME compliant data are available in the ArrayExpress database (E-MTAB-8764). Gene set enrichment analysis (GSEA) was performed using the Fisher’s exact test. Only the terms with GSEA *p*-value < 0.00001, a minimal overlap of twenty genes and an odds ratio > 2 were considered to be statistically significant.

### 3.6. Statistical Analysis

All of the statistical analyses were performed using GenEx 6, SAS release 9.4 (SAS Inc., Cary, NC, USA), and SPSS 25 (IBM Corporation, Armonk, NY, USA), unless stated otherwise. The expression of miRNAs in neoplastic and normal cells was compared by a Mann–Whitney test. All of the tested hypotheses were two-sided. As multiple tests were performed, we adjusted the *p*-values using the false discovery rate approach (FDR). The significance level was selected as alpha = 0.05; therefore, adjusted *p*-values below 0.05 were considered statistically significant unless otherwise stated.

## 4. Conclusions

Cultured normal human fibroblasts, pancreatic cancer-associated fibroblasts, and malignant cells from PDAC strongly express miR-21 and miR-210 and downregulate the expression of miR-217. These features are maintained even after being extended in vitro propagation. The expression of miR-21 and miR-210 is associated with high expression of IL-6 and molecules participating in the organization of the extracellular matrix. MiR-21, miR-210, and miR-217 seem to be important for the formation of a cancer-cell-supporting microenvironment. Moreover, these miRNAs may have some clinical relevance. There is increasing evidence supporting these miRNAs as reliable markers for PDAC diagnosis and prognosis.

## Figures and Tables

**Figure 1 ijms-24-03617-f001:**
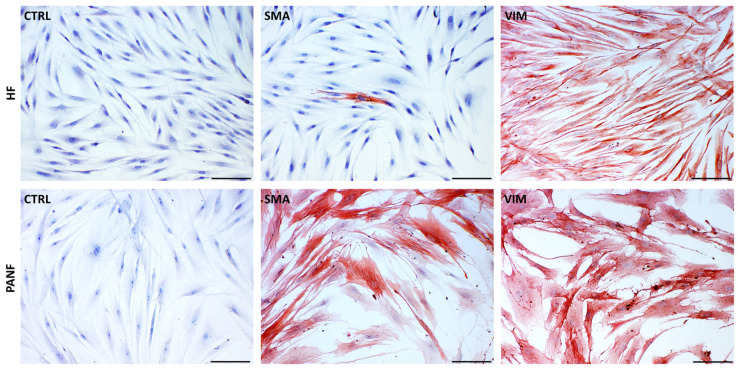
Detection of α-smooth muscle actin (SMA) and vimentin (VIM) in normal human dermal fibroblasts (HFs) and CAFs isolated from pancreatic ductal adenocarcinoma (PANF). Negative controls (CTRL) are also included. The bar is 100 μm.

**Figure 2 ijms-24-03617-f002:**
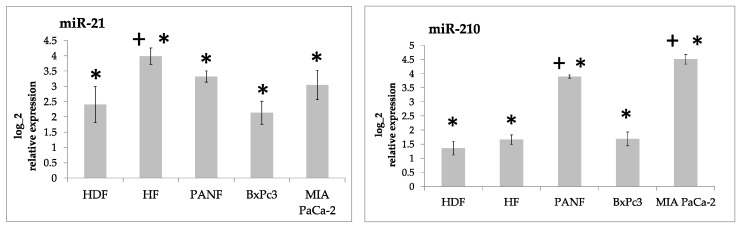
Log2 relative expression intensity of miR-21 and miR-210 in cultured human fibroblasts (HDF and HFs), cancer-associated fibroblasts prepared from pancreatic ductal adenocarcinoma (PANF) and cell lines prepared from this type of tumor (BxPC3 and MIA PaCa-2) compared to the tissue of the normal pancreas. Significant differences between cultured cells and the tissue of the normal pancreas, i.e., two-fold increase at adjusted *p*-value < 0.05, are marked by asterisks. Statistically significant differences between HDF of ectomesenchymal origin and other cell lines at adjusted *p*-value < 0.05 are marked by crosses.

**Figure 3 ijms-24-03617-f003:**
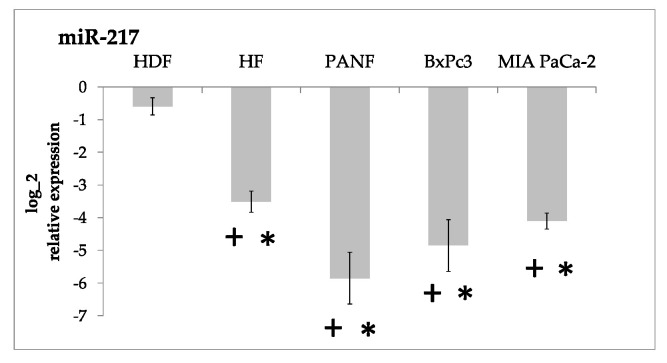
Downregulation of miR-217 in cultured human fibroblasts (HDF and HFs), pancreatic-ductal-cancer-associated fibroblasts (PANF), and PDAC cell lines (BxPC3 and MIA PaCa-2) compared to the tissue of the normal pancreas. Significant differences between cultured cells and the tissue of the normal pancreas, i.e., two-fold decrease at adjusted *p*-value < 0.05, are marked by asterisks. Statistically significant differences between HDF of ectomesenchymal origin and other cell lines at adjusted *p*-value < 0.05 are marked by crosses.

**Figure 4 ijms-24-03617-f004:**
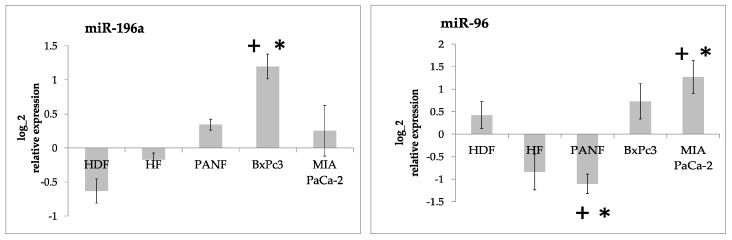
Log2 relative expression intensity of miR-196a and mir-96 in cultured human fibroblasts (HDF and HFs), cancer-associated fibroblasts prepared from pancreatic ductal adenocarcinoma (PANF), and cell lines prepared from this type of tumor (BxPC3 and MIA PaCa-2) compared to the tissue of the normal pancreas. Biologically significant differences between cultured cells and the tissue of the normal pancreas (two-fold change with adjusted *p*-value < 0.05) are marked by asterisks. Statistically significant differences between HDF and other cell lines at adjusted *p*-value < 0.05 are marked by crosses.

**Figure 5 ijms-24-03617-f005:**
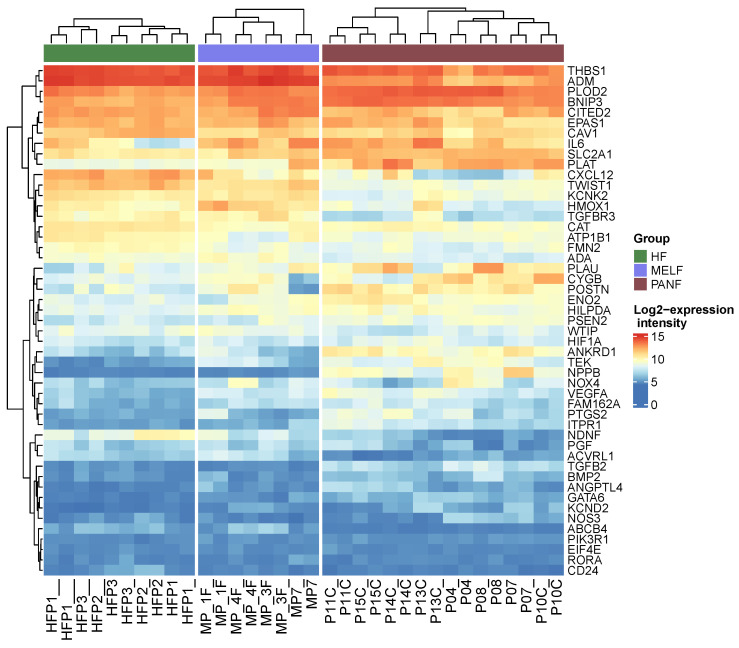
Heatmap demonstrating differential expression of genes influenced in hypoxia signalling (GO:0001666—response to hypoxia) in PANF, MELF CAFs, and HFs (normal fibroblasts). The gene encoding HIF-1α expression was increased by 60% in PANF in contrast to HFs, but not reaching statistical significance.

**Figure 6 ijms-24-03617-f006:**
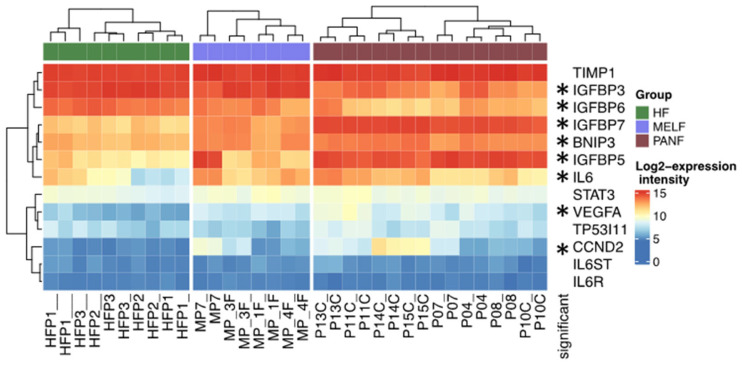
Expression of selected genes in normal fibroblasts (HFs), melanoma-associated fibroblasts (MELF), and cancer-associated fibroblasts prepared from pancreatic ductal adenocarcinoma (PANF). Statistically significant differences between PANF and HFs are marked by asterisks.

**Figure 7 ijms-24-03617-f007:**
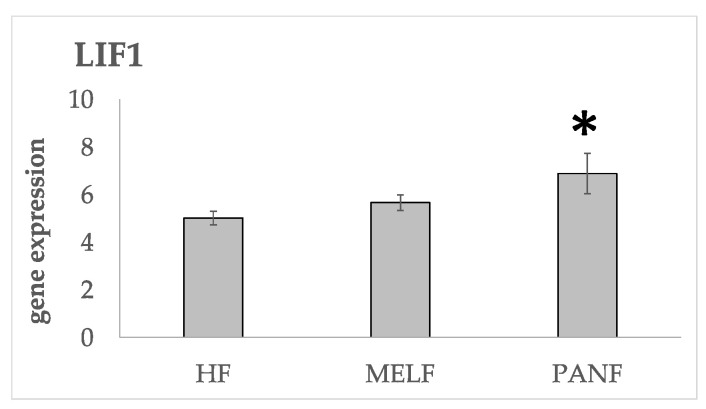
Comparison of the gene activity of LIF in normal fibroblasts and CAFs from the melanoma (MELF) and PDAC (PANF). A statistically significant difference between HFs and other cell lines at adjusted *p*-value < 0.05 is marked by an asterisk.

**Figure 8 ijms-24-03617-f008:**
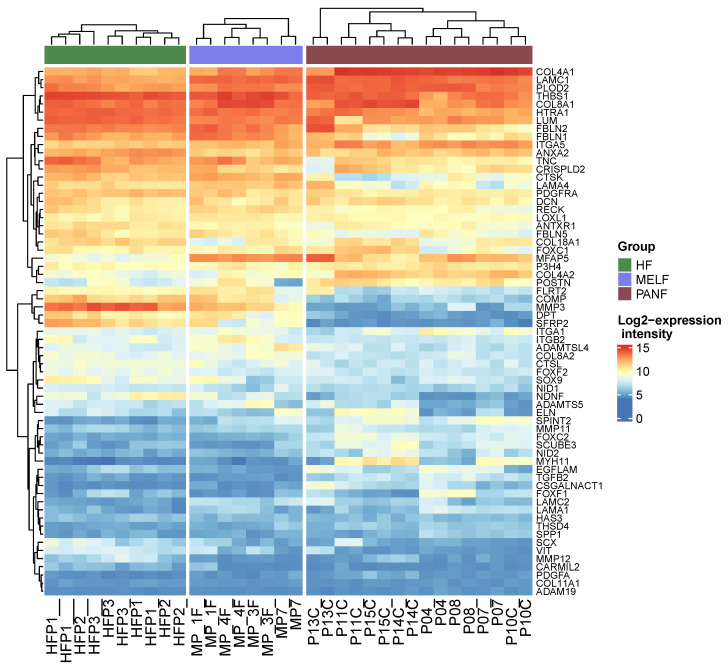
Heat map demonstrating differential expression of genes responsible for extracellular matrix organization in PANF and MELF compared to normal fibroblasts (GO term GO:0030198, extracellular matrix organisation).

**Table 1 ijms-24-03617-t001:** Role of miR-21 and miR-210 expression by CAFs/stellate cells in cancer progression.

Type of Cancer	Cancer-Supporting Effect of miR-21	Cancer-Supporting Effect of miR-210
Pancreatic ductal adenocarcinoma	[29,55,56,57,58,59,60,61,62]	[16,63]
Colorectal carcinoma	[58,64,65,66]	[67]
Lung adeno/non-small carcinoma	[68,69]	[70,71]
Breast	[72]	[73]
Gastric	[74]	Not available
Prostate	[75]	[76,77]
Thyroid medullary	[78]	Not available
Cutaneous malignant melanoma	Not reported	[79]

## Data Availability

All data supporting the reported results can be found in the institutions of authors or in public repositories.

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
