# Peer review of "Expression of Selected miRNAs in Normal and Cancer-Associated Fibroblasts and in BxPc3 and MIA PaCa-2 Cell Lines of Pancreatic Ductal Adenocarcinoma"

_ijms, 2023, doi:10.3390/ijms24043617_

Round 1

Reviewer 1 Report

The authors compare the expression of selected mRNAs in: normal fibroblasts, cancer-associated fibroblasts prepared from a ductal carcinoma, pancreatic carcinoma cell lines and a homogenate of paraffin-embedded sections from normal pancreatic tissue

The authors conclude: In cancer associated fibroblasts and cancer cell lines miR-21 and miR-210 were upregulated and miR-217 was downregulated compared to a tissue homogenate of normal pancreas. They discuss variations in miRs in relation to many factors including: hypoxia, stellate cells, Lif 1, extracellular matrix organization, leukemia inhibitory factor and IL-6.

This is an important study in the area of a dismal disease. Major objections, however, may be raised as to its design and presentation.

General comments

The procedures for obtaining the homogenates for normal pancreas is not outlined under Materials and Methods. The homogenates of normal pancreas includes also matrix cells like fibroblasts but also other types of cells like islet cells and stem cells.  Since the tissue was “a normal?” section of removed pancreas there was probably an infiltration of inflammatory cells also. These cells and there exosomes may influence the contents of miRs.

“Results and Discussion” may be better structured and needs a major revision.  The present outline is not entirely conclusive. Discussion should focus more on miRs. The discussions around  the results in Fig.5 , Fig.6, Table 1  and Fig. 8 have to be further outlined or omitted..

The use of abundant abbreviations is confusing for the reader and some are not clearly defined. A list of abbreviations would be a possible help.

Table of aberrations:

PDAC= pancreatic ductal adenocarcinoma

DACP ?

CAFs = cancer associated fibroblasts

PANF= cancer-associated fibroblasts prepared from pancreatic ductal carcinoma

PNAF?

MELF= cancer-associated fibroblasts from melanoma

HDF = cultured human fibroblasts

HF=     cultured normal human fibroblasts

HF = trunkal skin derived human fibroblasts

HDF fibroblasts = facial skin derived fibroblasts

In Introduction the authors say: ”we selected these five miRs because they play a significant role in PDAC”

·      What about other miRs not so well studied? What about network action of many miRs forming a landscape of nodes and edges? The up and downregulation of  single selected miRs may not give a definite clue to the aberrant function of the tumour microenvironment. A comment would be suitable.

Specific Comments

Page 3, line 119: “paraffin-embedded tissue from the normal pancreas” Please outline the procedures. Is there a risk that the composition of miRs  will be affected by the handling of the tissue in a way that differ from the cultured cell lines? This would make comparisons difficult.

Page 10, line 300. What does PNAF stand for?

Figure 2, Legend “compared to the tissue of the normal pancreas”. Can we rely on the miR data from the normal pancreas as prepared?

Figure 3, Legend “compared to the tissue of the normal pancreas”. Can we rely on the comparision?

Figure 4., Legend. “compared to the tissue of the normal pancreas”. Can we rely on the copmparision?

Figure 5. “Heatmap demonstrating..” Why is this expression of genes in hypoxia signaling included when the HIF1A gene is not upregulating HIF-1alpha? Explain or omit.

Figure 8.  Why is this included? Legend should be further outlined or the figure should be omitted.

Figure 9. The contents should be better explained or the figure should be omitted.

Reviewer 2 Report

In this manuacript the Authors studied the differential espression of some miRNAs known to be involved in pancreatic adenocarcinoma. Although the results seem interesting, they should be validated by analyzing the espression of genes known to be putative target of these miRNAs by immunohistochemistry on tumor tissue . Moreover the modulation of HIF was not explored by immunohistochemistry. Only by assessing the effects of differential espression of miRNAs on tissue in vivo, One should argue some conclusions. Otherwise, the research still remains speculative 

Reviewer 3 Report

In this study, authors evaluated expression levels of miR-21, -96, -196a, -210, and -217 in cancer associated fibroblasts (CAFs) prepared from pancreatic ductal carcinomas (PDAC), abbreviated as PANFs, in comparison to human dermal fibroblast (HF), PDAC cell lines and normal pancreatic tissue. The obtained results are further compared and discussed with gene expression levels detected in HFs and PANFs in recently published work (Novak, S., et al., 2021). 

Due the originality of the research and the importance of the obtained results, I have a few comments which I hope could help to improve the manuscript.

Major questions/concerns

-       Page 4-5, part 2.2 and page 12, part 3.1

It is not clear how many samples of HFs, HDFs, PANFs and normal pancreatic tissues were used for miRNA evaluation. If only one cell line/tissue of each sample type was used, what is hidden behind the error bars in Figures 2-4 and what types of replicates were used for statistic? This should be clearly stated in the Methods and in Figures 2-4.

-       Page 4-5, part 2.2

The authors compared the levels of miRNAs obtained from cultured cells in comparison to FFPE block of normal tissue. Due to the huge difference in RNA quality between these types of samples, it is usually not possible to combine them without first verifying the correlation. However, the information about the quality of the RNA (e.g., RINs) or about the previous correlation study is missing. The results should be verified using fresh frozen normal pancreatic tissue or (commercially available) cell line of normal pancreas.

-       Page 4-5, part 2.2

There is high variability in miRNA levels evaluated in normal fibroblasts (HDF, HF) and cancer cell lines (BxPc3, MIA Paca-2). Why only one biological replicate of PANF was used? In the case of mRNA analysis, around 16 samples of PANF were analyzed. The number of PANF samples should be increased for a better idea of the variability of miRNA expression in this type of samples.

-       Page 13, part 3.4

Which and how many reference miRNAs were used to calculate ΔΔCt and how were selected?

Minor comments

-       Page 2, line 65-66

“The cancer-supporting properties of CAFs are significantly stimulated by exo- 65 somes produced by cancer cells containing numerous miRNAs as bioactive cargo [3].“

This should be supported by citations of original works.

-       Page 3, part 2.1

In immunohistochemical characterization of HF and PANF is discrepancy between expression of alpha-smooth muscle actin on protein and mRNA levels in PANF cells. How do the authors explain it?

-       Page 4-5, part 2.2

“adjusted p-value” is given in results of miRNA analysis (e.g., line 139) but FDR correction was done only for mRNA analysis according to authors. And similarly, the definition of the statistical significance should be unified in case of mRNA analysis, e.g., “q-value of < 0.05” line 410 vs “p-value of < 0.05” line 21 in Supplement.

-       Figure 5 and 8

It would be more practical to list the names of all genes as in Figure 6.

-       Some of the sentences or passages, mainly in Results and Discussion, are written unclearly or imprecisely, e.g.,

line 170

“deregulation of miR-217 may predict the cancer presence in PDAC patiens [34]” – What type of deregulation? In what type of samples?

line 316-318

“PANF isolated from PDAC maintained their high levels of both miR-21 and -210 even after prolonged in vitro propagation. These data show that CAFs of PDAC participate in elevated levels in biological fluids.” - What data do the authors refer to?

Round 2

Reviewer 1 Report

The manuscript has been revised according to my comments and the questions raised have been mainly answered.

One weakness is that the tissue control material is from one single patient which should be mentioned. For further information on its composition in the Methods reference 14 should be added.

On page 6, line 220: "the expression of the HIF1A gene was clearly upregulated  in PANF cells, yet it did not reach statistic significance" .

If there is no statistical significance there is no clear upregulation. The authors have to modify the text.

Further the text needs some further linguistic revision
